# You Never Stop Dancing: Non-freezing Dance Generation via Bank-constrained Manifold Projection

**Jiangxin Sun**[*]
Sun Yat-sen University
sunjx5@mail2.sysu.edu.cn

**Chunyu Wang**
Microsoft Research Asia
chnuwa@microsoft.com

**Huang Hu**
Peking University
tonyhu@pku.edu.cn

**Hanjiang Lai**
Sun Yat-sen University
laihanj3@mail.sysu.edu.cn

**Zhi Jin**
Sun Yat-sen University
jinzh26@mail.sysu.edu.cn

**Jian-Fang Hu**[†]
Sun Yat-sen University
hujf5@mail.sysu.edu.cn

## Abstract

One of the most overlooked challenges in dance generation is that the auto-regressive frameworks are prone to freezing motions due to noise accumulation. In this paper, we present two modules that can be plugged into the existing models to enable them to generate non-freezing and high fidelity dances. Since the high-dimensional motion data are easily swamped by noise, we propose to learn a low-dimensional manifold representation by *an auto-encoder with a bank of latent codes*, which can be used to reduce the noise in the predicted motions, thus preventing from freezing. We further extend the bank to provide explicit priors about the future motions to disambiguate motion prediction, which helps the predictors to generate motions with larger magnitude and higher fidelity than possible before. Extensive experiments on AIST++, a public large-scale 3D dance motion benchmark, demonstrate that our method notably outperforms the baselines in terms of quality, diversity and time length.

## 1 Introduction

Dancing to music has been one of the most popular art forms since ancient days. It can vividly express human's emotions and fulfill social communications even before symbolic languages came along. Nowadays, there are a booming number of people sharing their dance videos on the popular media platforms such as YouTube and TikTok, which drives the strong need for automatic AI choreography to help users create their own dances. The task is related to general human motion prediction [22, 25, 31, 38, 39, 42] except that it poses new challenges: i) dance generation needs to produce high-fidelity motions for about three minutes to cover a music which is much longer than that in general motion prediction; ii) dance generation needs to handle more diverse and stylistic motions (*e.g.*, ballet Jazz, hip-pop, *etc.*). The high spatio-temporal complexity requires more expressive models so as to generate high-fidelity dance motions.

---

[*]This work was done when Jiangxin Sun was an intern at Microsoft Research Asia.

[†]Hu is also with Guangdong Province Key Laboratory of Information Security Technology, Guangzhou, China and Key Laboratory of Machine Intelligence and Advanced Computing, Ministry of Education, China.

36th Conference on Neural Information Processing Systems (NeurIPS 2022).

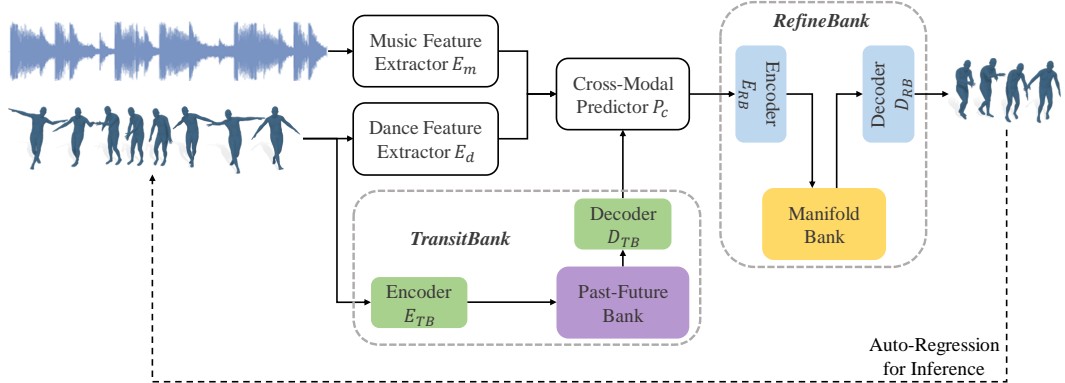

Figure 1: The motion prediction framework. It first uses the past motions to query TransitBank to retrieve the priors about the future motions, which are fed to the predictor to get high-fidelity motion predictions. Then RefineBank refines the predictions to reduce error accumulation.

The state-of-the-art methods follow a cross-modal prediction framework where the future motions are predicted based on the past motions and music in an auto-regressive way [11, 16, 20, 30]. However, the generated motions are prone to freezing and converging to small-magnitude motions after only several seconds. The main reason is that the prediction error will accumulate in the generation process, and eventually can not be handled by the neural predictors. Besides, motion prediction suffers from the huge uncertainty and ambiguity because of the high spatio-temporal complexity of the task. As a result, the models tend to predict mean poses [42] instead of large-magnitude and high-fidelity motions if there are not informative priors about the future motions.

In this paper, we present two modules that can be plugged into the existing auto-regressive models to achieve non-freezing large-magnitude motion generation. Figure 1 illustrates an overview. Firstly, to prevent from error accumulation, we present *RefineBank* to learn a low-dimensional manifold representation for the high-dimensional motion data. It equips an auto-encoder with a bank of latent codes to *tightly* constrain the manifold to be close to the ground-truth (GT) motions and meanwhile far from the ones with noise. This representation allows us to remove the noise in the predicted motions by projecting them to the learned manifold as illustrated in Figure 2. With RefineBank, the baseline method [20] can already generate full choreography for complete musics in the dataset without freezing motions. Secondly, inspired by the fact that most dances can be coarsely constructed by a number of basic motion segments, we present *TransitBank* to learn and memorize the frequently used <past, future> motion dynamics on top of the manifold. Given past motions, it provides explicit priors about the future motions to reduce the uncertainty and ambiguity in prediction, which can effectively facilitate the higher-fidelity dance motion generation with larger magnitude than before.

We conduct extensive experiments to evaluate our approach on the AIST++ dataset [20]. Not only does our method notably outperform the baselines on the existing metrics, but also shows better results on our newly introduced freezing rate metric. In addition, the user study indicates that people have obvious preferences toward the dances generated by our method. Our contributions are summarized as follows: 1) This is the first time we see evidence that the prediction-based methods can generate long-term dance motions on AIST++, which paves the way for full choreography for entire music; 2) We present *RefineBank* and *TransitBank* which can be plugged into most motion-prediction based methods to achieve long-term non-freezing dance generation; 3) We introduce new metrics that can quantitatively evaluate the freezing situations in the motion sequences.

## 2 Related Work

**Prediction-based Methods**   This line of work treats dance generation as a motion prediction problem and has achieved promising results. A number of network architectures have been proposed including CNNs [9, 16], RNNs [1, 11, 32, 35, 41], GCNs [4, 26, 36], GAN [30] and Transformers [12, 18–20]. Several works focus on the alignment of motion and music. For instance, Sun et al. [30] and Li et al. [18] use a classifier to test the authenticity of the predicted motion conditioned on the

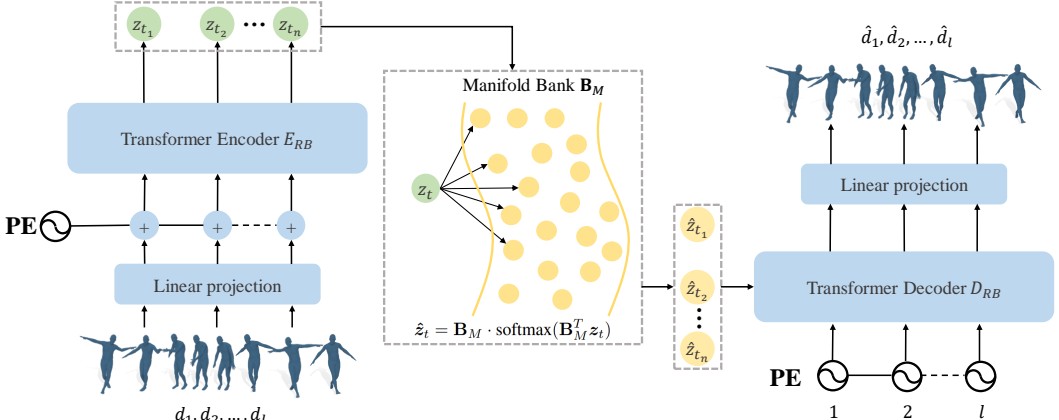

Figure 2: **RefineBank** takes a motion sequence with $l$ frames as input, projects them to a low-dimensional manifold, and reconstructs the sequence with minimum noise.

music. Zhang et al. [41] and Huang et al. [12] learn the dance style embeddings to provide prior information to the predictor. Few works have studied the freezing problem. Huang et al. [11] propose a curriculum learning strategy to bridge the gap between training and inference by alternately feeding predicted and GT motions to the predictor. Li et al. [20] present future-n full-attention to replace the traditional shift-by-1 casual-attention to leverage the temporal context. However, the predicted motions are still prone to freezing after several seconds. Our approach is also a prediction-based method. But different from the previous methods, we present two plug-in modules that can achieve longer-term non-freezing motion generation.

**Retrieval-based Methods**   Some earlier works compose a complete dance by retrieval [14,24,27,28]. They select the closest predefined motion segments in a pre-built database based on music, and construct a sequence with the proper transition routines. Lee et al. [17] and Ye et al. [37] use deep networks to generate future motion segments from the input music and past motion segments. Duan et al. [3] propose an attention-based MLP to translate all music phrases to motion segments. Chen et al. [2] propose to predict dance sequences through traversing the node transition routines on a motion graph and introduce the choreography-oriented constraints to compose the final dance motion sequences. However, they cannot generate new motions beyond the database. Our approach is also inspired by the retrieval-based methods in that we construct a bank/database to learn common motion dynamics for the predictor. The core difference is that the bank is automatically learned from data without manual efforts and our approach can generate new motions beyond the database.

## 3   Preliminaries

Given the past and future music features $\boldsymbol{m}_{1:t} \parallel \boldsymbol{m}_{t+1:t+K}$, and the past dance motions $\hat{\boldsymbol{d}}_{1:t}$, the task aims to predict the future motions $\hat{\boldsymbol{d}}_{t+1:t+K}$. The input music features $\boldsymbol{m}_t \in \mathbb{R}^{35}$ are obtained from Librosa [23] consisting of 1-dim envelope, 20-dim MFCC, 12-dim chroma, 1-dim one-hot peaks and 1-dim one-hot beats. The motions are represented by SMPL [21] model, which consists of parameters referring to body shape, human pose and translation. The shape parameters are dance-irrelevant values which mainly capture the expansion/shrink of human body such as taller or shorter, we only consider the prediction of parameters respecting to human pose and global translation in $\hat{\boldsymbol{d}}_t \in \mathbb{R}^{219}$. We follow the state-of-the-art dance generation architecture [20] as shown in Figure 1. It has a music feature extractor $E_m$, a dance feature extractor $E_d$ and a cross-modal predictor $P_c$. The auto-regressive prediction process can be formulated as:

$$\hat{\boldsymbol{d}}_{t+1:t+K} = P_c(E_m(\boldsymbol{m}_{1:t+K}), E_d(\hat{\boldsymbol{d}}_{1:t})). \tag{1}$$

Although previous works such as [11, 20] have attempted to improve the long-term generation performance, the predicted motions are still prone to freezing after only several seconds.

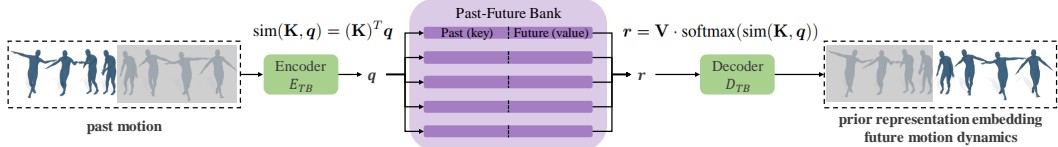

Figure 3: **TransitBank**: Given first half of the motion sequence, it produces a prior representation for the second half of the sequence by computing the weighted average of the values of the bank items to reduce the uncertainty and ambiguity in motion prediction.

## 4 Method

The dance motions are believed to lie on a low-dimensional manifold since the body parts are highly correlated [10, 29]. We take advantage of the nice property and present *RefineBank (RB)* to reduce the noise in the motions. It learns an auto-encoder with a bank of latent codes to tightly represent the compact motion manifold. By projecting noisy or corrupted motions onto the manifold, we can remove the noise in the motions, which prevents error accumulation in the auto-regressive generation. Secondly, we propose *TransitBank (TB)* on top of the learned manifold, which maintains a past-future motion dynamics bank to provide explicit priors about the future motions. The priors narrow down the motion prediction space which facilitates high-fidelity motion generation with large magnitude. Mathematically, the prediction process can be formulated as:

$$\hat{\boldsymbol{d}}_{t+1:t+K} = \text{RB}(\, P_c(\, E_m(\boldsymbol{m}_{1:t+K}),\, E_d(\hat{\boldsymbol{d}}_{1:t}),\, \text{TB}(\hat{\boldsymbol{d}}_{1:t})\,)\,). \tag{2}$$

### 4.1 RefineBank

Figure 2 shows the components of RefineBank which has an encoder $E_{RB}$, a decoder $D_{RB}$, and a manifold bank $\mathbf{B}_M \in \mathbb{R}^{C \times N}$. The bank $\mathbf{B}_M$ has $N$ learnable latent codes of dimension $C$ which span the low-dimensional dance manifold. As will be described later, the bank is learned from the GT motions so it can be interpreted as a prior probabilistic distribution where the motion data with noise will have the small likelihood.

For a (noisy) motion sequence $\hat{\boldsymbol{d}}_{1:l}$ produced by the predictor, we first transform each of them to a latent feature by a transformer-based encoder $E_{RB}$. Concretely, we uniformly sample $n$ features at different time steps $\boldsymbol{z} = \left\{\boldsymbol{z}_t | t \in \left\{0, \frac{l}{n-1}, \frac{2l}{n-1}, ..., l\right\}\right\}$ to represent the sequence. Then we project each $\boldsymbol{z}_t$ to the manifold represented by the bank to remove the prediction error. Specifically, for each latent feature $\boldsymbol{z}_t$, we compute the similarity scores between $\boldsymbol{z}_t$ and the latent codes in $\mathbf{B}_M$, and use the learnable weights to project $\boldsymbol{z}_t$ onto the manifold to get $\hat{\boldsymbol{z}}_t$:

$$\hat{\boldsymbol{z}}_t = \mathbf{B}_M \cdot \text{softmax}(\mathbf{B}_M^T \boldsymbol{z}_t). \tag{3}$$

The projected latent features $\{\hat{\boldsymbol{z}}_t\}$ will be fed to the transformer-based decoder $D_{RB}$ to generate the motions that we expect to have little noise.

Compared with previous bank-based autoencoder approaches for image generation such as VQ-VAE [33], we have two unique designs in our RefineBank for dance generation. The first is to use soft-assignment instead of hard nearest-neighbor assignment to prevent simply repeating existing dance movements and improve the diversity of the generated dances. The second is that we follow the multi-head implementation in transformers to encode the input motion sequence with several different latent features such that more details can be preserved.

### 4.2 TransitBank

The high spatio-temporal complexity of the motion space increases the uncertainty and ambiguity of motion prediction. Hence, the motion predictors are prone to generating small-magnitude motions. To address the problem, we present TransitBank which is inspired by the fact that there exist a number of basic short motion segments that are frequently used in many dances. TransitBank aims to exploit such cues to estimate the prior distributions for future motion prediction.

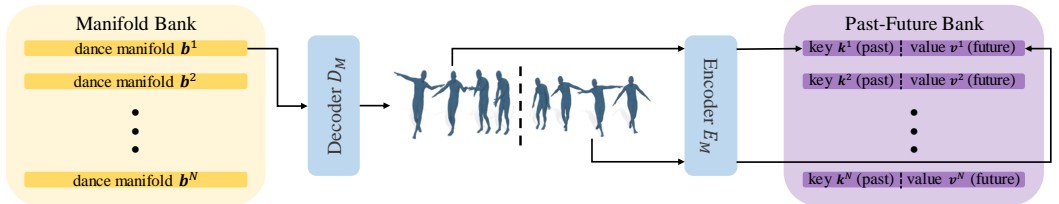

Figure 4: Constructing the past-future bank from the manifold bank.

Concretely, TransitBank works by dividing a motion sequence into past and future sub-sequences and storing them as past-future pairs. Then given the past motions as input, it queries the bank and reads the corresponding future motion dynamics from the bank. This prior information is fed to the cross-modal predictor to reduce the uncertainty and ambiguity in motion prediction.

Figure 3 shows the structure of TransitBank. It consists of an query encoder $E_{TB}$, a past-future bank $B_{PF}$, and a read decoder $D_{TB}$. The past-future bank $\mathbf{B}_{PF} = \left\{ (\mathbf{K}, \mathbf{V}) | \mathbf{K}, \mathbf{V} \in \mathbb{R}^{C \times N} \right\}$ is directly constructed as the <past, future> extension of manifold bank $\mathbf{B}_M$. As for $E_{TB}$ and $D_{TB}$, we use a simple transformer structure for encoder and decoder, respectively. To avoid information leakage in case that the cross-modal predictor directly copies the motions in TransitBank, we design the query-read process by using the attention mechanism rather than finding closest one which will effectively blur the future motions. Specifically, we encode input motion $\boldsymbol{d}_{1:l}$ via $E_{TB}$ and take the average pooling of all outputs as the query vector $\boldsymbol{q}$. Then we compute the similarity between $\boldsymbol{q}$ and key features $\mathbf{K}$ following:

$$\text{sim}(\mathbf{K}, \boldsymbol{q}) = (\mathbf{K})^T \boldsymbol{q}. \tag{4}$$

In the read process, we compute the future motion prior $\boldsymbol{r}$ as the weighted average of the corresponding values $\mathbf{V}$:

$$\boldsymbol{r} = \mathbf{V} \cdot \text{softmax}(\text{sim}(\mathbf{K}, \boldsymbol{q})). \tag{5}$$

Finally, we decode the vector $\boldsymbol{r}$ via $D_{TB}$ and feed it to the cross-modal predictor as additional tokens.

### 4.3 Model Learning

This section describes how we learn the two banks from training data. It is worth noting that the two banks are not independent. Instead, TransitBank is a <past, future> extension of RefineBank (*i.e.*, mapping motion prefixes to suffixes). In the following part, we first illustrate how we learn the bank in RefineBank and then describe how to derive the one in TransitBank. Finally, we describe the training optimization strategy.

**Bank in RefineBank**   We first initialize the manifold bank $\mathbf{B}_M$ by clustering to obtain a proper initialization where items are representative and widely used in GT motion. Specifically, we obtain a number of motion segments by applying a sliding window with length $l$ to all training motion sequences. Then AP clustering [5] is used to obtain a number of cluster centers from the segments. We apply a transformer-based encoder $E_M$ to encode each cluster center $\boldsymbol{d}_{1:l}^i$ and compute the average of the encoding outputs of the $l$ frames to initialize one bank elements $\boldsymbol{b}^i \in \mathbf{B}_M$.

After initialization, we train the encoder $E_M$, decoder $D_M$ and the bank $\mathbf{B}_M$ to reconstruct all motion segments $\boldsymbol{d}_{1:l}$ using the method described in Section 4.1. The only difference is that we project the latent feature $\boldsymbol{z}$ to its closest item $\boldsymbol{b}$ of $\mathbf{B}_M$ as:

$$\hat{\boldsymbol{z}} = \arg \min_{\boldsymbol{b} \in \mathbf{B}_M} \|\boldsymbol{z} - \boldsymbol{b}\|. \tag{6}$$

**Bank in TransitBank**   Constructing the past-future bank $\mathbf{B}_{PF}$ from the manifold bank $\mathbf{B}_M$ is straightforward. As shown in Figure 4, for each element $\boldsymbol{b} \in \mathbf{B}_M$, we decode it to a motion sequence

$\hat{\boldsymbol{d}}_{1:l}$ via the learned transformer decoder $D_M$. Then we divide the sequence into two parts: $\hat{\boldsymbol{d}}_{1:l/2}$ and $\hat{\boldsymbol{d}}_{l/2+1:l}$, which are fed to the encoder $E_M$ to obtain the latent codes representing the past (**K**) and future (**V**), respectively. In that way, we obtain $\mathbf{B}_{PF} = \left\{(\mathbf{K}, \mathbf{V}) | \mathbf{K}, \mathbf{V} \in \mathbb{R}^{C \times N}\right\}$.

**Optimization Strategy**  We train our model in three stages to ensure that the learned manifold bank can accurately reconstruct the GT motions. In the first stage, we just follow the implementation in VQ-VAE [33] and optimize the manifold bank by minimizing the following loss:

$$\mathcal{L}_{\text{ManifoldBank}} = \left\| \hat{\boldsymbol{d}}_{1:l} - \boldsymbol{d}_{1:l} \right\|_2^2 + \|\text{sg}[\boldsymbol{z}] - \hat{\boldsymbol{z}}\|_2^2 + \beta \|\boldsymbol{z} - \text{sg}[\hat{\boldsymbol{z}}]\|_2^2. \tag{7}$$

The first term minimizes the reconstruction error. The second part is the "item loss" [33] to update items in manifold bank, where sg denotes "stop gradient". This objective function moves the items close to the outputs of the encoder. The sg[·] operator is implemented by the identification function during forward computation with zero partial derivatives. The third part is "commitment loss" [33] to ensure the output of encoder commits to an item and the value of $\beta$ is set as 0.2, empirically. We optimize the encoder with the first and third loss terms. The bank items are updated with only the second loss term and the decoder is trained with only the first term. Once we finalize the manifold bank, we can compute the past-future bank. Note that we do not update these two banks in the following stages.

In the second stage, we train the encoder and decoders in RefineBank to reconstruct all motion sequences of the training set with only reconstruction loss.

$$\mathcal{L}_{\text{RefineBank}} = \left\| \hat{\boldsymbol{d}}_{1:l} - \boldsymbol{d}_{1:l} \right\|_2^2, \tag{8}$$

In the third stage, we train the whole framework in an end-to-end manner with the L2 loss between predicted motion and GT motion.

$$\mathcal{L}_{\text{Prediction}} = \left\| \hat{\boldsymbol{d}}_{t+1:t+K} - \boldsymbol{d}_{t+1:t+K} \right\|_2^2, \tag{9}$$

# 5 Experiments

## 5.1 Experimental Settings

**Dataset**  We evaluate our method on the largest AIST++ [20] dance dataset that contains 60 music pieces belonging to 10 dance genres. In total, there are 992 3D pose sequences at 60 FPS. Following [20], we use 952 samples for training and the rest 40 for evaluation.

**Implementation Details**  The model takes 240 frames of music and 120 frames of motions as input and predicts the next $K = 20$ frames of motions. We use the same network structures for the music extractor, motion extractor and cross-modal predictor as FACT [20]. For the encoders and decoders in RefineBank and TransitBank, we use transformers with 4 layers and 10 attention heads with 2048 hidden size. The number of items in manifold bank and past-future bank is 256 and each item is a 2048-dim latent vector. The input to RefineBank and TransitBank is a motion sequence with $l = 120$ frames. Since $K < l$, we concatenate the previous $l - K$ frames of motions with the predictor output to feed to RefineBank. We construct a separate manifold bank and past-future bank for each dance genre. In the first training stage, we adopt Adam optimizer [15] with a learning rate of $1 \times 10^{-4}$ to train the manifold bank for 50 epochs. In the second training stage, we pre-train the RefineBank using Adam optimizer with a learning rate of $1 \times 10^{-4}$ for 25 epochs. In the third stage, we train the whole framework with Adam optimizer for 150 epochs. The learning rate starts with $1 \times 10^{-4}$ and decreases to $\left\{1 \times 10^{-5}, 1 \times 10^{-6}\right\}$ after $\{30, 90\}$ epochs, respectively. The whole training process takes about four days on four NVIDIA GeForce RTX 2080Ti GPUs.

**Metrics**  The previous works mainly evaluate the dance generation results from three aspects: quality, diversity and alignment. Following [20], we compute FID [8] (Frechet Inception Distances) on the kinetic features (denoted as $\mathbf{FID}_k$) and geometric features (denoted as $\mathbf{FID}_g$), respectively, to measure quality. We use the fairmotion toolbox [6] to extract the features. For diversity, we compute the average Euclidean distance in the kinetic (denoted as $\mathbf{Dist}_k$) and geometric (denoted as $\mathbf{Dist}_g$)

Table 1: Comparison to the state-of-the-art methods on the AIST++ dataset.

| Method | Quality | | | | | Diversity | | Align | User Study |
| | FID$_k$ ↓ | FID$_g$ ↓ | $\Delta_{\text{Pose}}$ ↑ | $\Delta_{\text{Trans}}$ ↑ | Freeze ↓ | Dist$_k$ ↑ | Dist$_g$ ↑ | BeatAlign ↑ | Win Rate ↑ |
|---|---|---|---|---|---|---|---|---|---|
| GT | - | - | 3.28 | 1.16 | 18.7% | 9.06 | 7.31 | 0.292 | 31.7% |
| Li et al. [19] | 86.43 | 43.46 | 1.02 | - | 59.0% | 6.85 | 3.32 | 0.232 | 95.8% |
| DanceNet [43] | 69.18 | 25.49 | 1.25 | 0.80 | 46.8% | 2.86 | 2.85 | 0.232 | 90.8% |
| Revolution [11] | 73.42 | 25.92 | - | - | - | 3.52 | 4.87 | 0.220 | 84.2% |
| FACT [20] | 35.35 | 22.11 | 1.33 | 1.07 | 39.0% | 5.94 | 6.18 | 0.241 | 86.7% |
| Ours | **25.96** | **13.42** | **1.64** | **1.36** | **29.6%** | **7.68** | **6.59** | **0.249** | - |

Table 2: Ablation study of RefineBank and TransitBank.

| Method | Quality | | | | | Diversity | | Alignment |
| | FID$_k$ ↓ | FID$_g$ ↓ | $\Delta_{\text{Pose}}$ ↑ | $\Delta_{\text{Trans}}$ ↑ | Freezing ↓ | Dist$_k$ ↑ | Dist$_g$ ↑ | BeatAlign ↑ |
|---|---|---|---|---|---|---|---|---|
| Baseline | 35.35 | 22.11 | 1.33 | 1.07 | 39.0% | 5.94 | 6.18 | 0.241 |
| + RefineBank | 28.67 | 16.38 | 1.53 | 1.15 | 32.1% | 6.65 | 6.34 | 0.246 |
| + TransitBank | 31.24 | 19.18 | 1.49 | 1.31 | 34.3% | 7.42 | 6.47 | 0.245 |
| Ours | **25.96** | **13.42** | **1.64** | **1.36** | **29.6%** | **7.68** | **6.59** | **0.249** |

feature space across the generated motions. For dance-music alignment, we adopt **Beat Alignment Score** in [20] to compute average distance between every kinematic beat and its nearest music beat. Since the freezing problem is largely overlooked previously, there are no metrics available to evaluate. We propose to compute the average values of the temporal differences of the pose and translation parameters in the whole sequence, which are termed as $\Delta_{\text{Pose}}$ and $\Delta_{\text{Trans}}$, respectively. In addition, we also calculate the **Freezing Rate** of each sequence. We divide a sequence into non-overlapping sub-sequences of 60 frames, and for each sub-sequence, if $\Delta_{\text{Pose}} \leq \Delta_{\text{Pose}}^{\text{gt}}$ and $\Delta_{\text{Trans}} \leq \Delta_{\text{Trans}}^{\text{gt}}$ where $\Delta^{\text{gt}}$ is a predefined threshold statistically derived from the training set, we regard it as a freezing sub-sequence. Then we compute the percentage of freezing sub-sequences.

## 5.2 Comparison to the State-of-the-arts

We compare our approach to a number of recent methods including Li et al. [19], Dancenet [43], DanceRevolution [11], and FACT [20]. Our approach employs the same structure as FACT except that it has RefineBank and TransitBank. For each music, we generate a motion sequence with 1200 frames (20 seconds). The experiment results are shown in Table 1. Since DanceRevolution [11] predicts 3D keypoint positions, we cannot compute the SMPL-based freezing metrics. The approach in [19] does not predict the translation parameters. As shown, our approach outperforms the state-of-the-art methods on all metrics. Interestingly, the $\Delta_{\text{Trans}}$ of our method is even larger than that of GT. We visually compare our generated motions and GT motions and find that it is because GT often have stationary poses at transition moments. By contrast, it seems difficult for learning-based methods to predict stationary poses.

We present a detailed analysis for one sequence in Figure 5. As we can see, the motion and translation changes of FACT gradually decrease to a small number suggesting that the freezing situation occurs. In contrast, the numbers of our method are always above those of FACT and do not freeze. The results validate that our proposed bank-based manifold projection and past-future dynamic priors indeed improve the quality of the generated motions.

**User Study** We conduct a user study to investigate how people think of the dances generated by our method and the other ones. We invite 20 participants and each participant is asked to watch 30 pairs of comparison videos. Each pair consists of our dance and one competitor's generated with the same music. We ask each participant to determine "which person is dancing better to the music" and provide the statistics in Table 1 (last column). Our approach significantly outperforms the other methods in user study. We can keep at least 84.2% win rate to the SOTA methods including DanceRevolution and FACT. Notably, we can achieve 31.7% win rate compared to GT motions. We

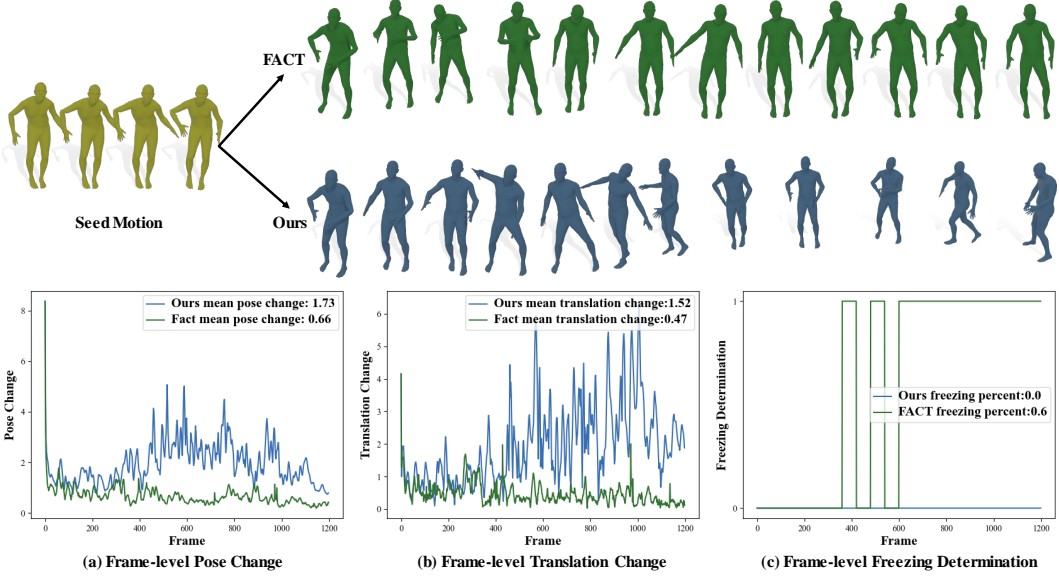

Figure 5: The top row shows the motions generated by FACT and our method for the same music. The bottom provides frame-level statistics for the two sequences. In figure (c), 0 represents non-freezing and 1 means freezing. Best viewed in color.

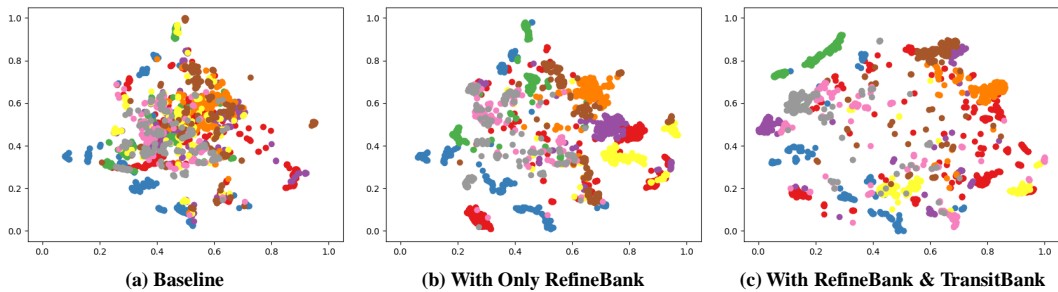

Figure 6: t-SNE visualization of the generated dances. Each dot represents a 3D pose and different colors represent different genres of the music used to generate the poses. Best viewed in color.

also collect detailed feed-backs and find that our generated dance is generally thought to be more diverse and non-freezing. The main problem with FACT is that the motions freeze frequently while the problem with DanceRevolution is that the motions are unnatural. Compared to GT motions, ours are thought to lack suitable transition motions and precise beat alignment which is a general problem faced by most prediction-based methods.

## 5.3 Ablation Study

**Ablation Study of RefineBank and TransitBank**  The experiment results are shown in Table 2. Our first observation is that adding RefineBank to the baseline notably improves $\Delta_{\text{Pose}}$ meaning that the method reduces the chance of getting freezing motions. Meanwhile, the improvement also brings benefits to dance quality and diversity. However, the variation of the global positions of the dancers, $i.e.$, $\Delta_{\text{Trans}}$, is only slightly improved. This is expected since RefineBank only guarantees that the refined motions are on the manifold. In contrast, adding TransitBank notably improves $\Delta_{\text{Trans}}$. Meanwhile the diversity metrics are also notably improved. The results suggest that by exploiting the motion dynamic priors the method can predict high-fidelity and diverse motions with large magnitude instead of mean poses. Finally, the two modules are complementary to each other and combining them will further improve the results on all metrics.

Table 3: Comparison of the bank-based AE and other options.

| Method | Quality | | | | | Diversity | | Align |
|---|---|---|---|---|---|---|---|---|
| | $\text{FID}_k \downarrow$ | $\text{FID}_g \downarrow$ | $\Delta_{\text{Pose}} \uparrow$ | $\Delta_{\text{Trans}} \uparrow$ | Freezing $\downarrow$ | $\text{Dist}_k \uparrow$ | $\text{Dist}_g \uparrow$ | BeatAlign $\uparrow$ |
| AE | 29.85 | 17.64 | 1.52 | 1.32 | 33.5% | 7.48 | 6.49 | 0.245 |
| VAE | 29.67 | 17.05 | 1.53 | 1.32 | 33.1% | 7.49 | 6.50 | 0.245 |
| Bank-AE (Discrete) | 27.48 | 15.19 | 1.60 | 1.34 | 30.4% | 7.45 | 6.48 | 0.247 |
| Bank-AE (Ours) | **25.96** | **13.42** | **1.64** | **1.36** | **29.6%** | **7.68** | **6.59** | **0.249** |

Table 4: Evaluation on the number of latent features.

| | Quality | | | | | Diversity | | Alignment |
|---|---|---|---|---|---|---|---|---|
| | $\text{FID}_k \downarrow$ | $\text{FID}_g \downarrow$ | $\Delta_{\text{Pose}} \uparrow$ | $\Delta_{\text{Trans}} \uparrow$ | Freezing $\downarrow$ | $\text{Dist}_k \uparrow$ | $\text{Dist}_g \uparrow$ | BeatAlign $\uparrow$ |
| n=1 | 27.94 | 15.61 | 1.59 | 1.34 | 31.2% | 7.51 | 6.52 | 0.246 |
| n=2 | 26.72 | 14.78 | 1.62 | 1.35 | 30.6% | 7.63 | 6.57 | 0.248 |
| n=3 | 26.15 | 13.95 | 1.64 | 1.36 | 29.9% | 7.66 | 6.59 | 0.249 |
| n=4 | **25.96** | **13.42** | **1.64** | **1.36** | **29.6%** | 7.68 | 6.59 | **0.249** |
| n=5 | 26.67 | 13.88 | 1.63 | 1.35 | 30.1% | **7.72** | **6.63** | 0.248 |

Table 5: Evaluation on the number of bank items.

| | Quality | | | | | Diversity | | Alignment |
|---|---|---|---|---|---|---|---|---|
| | $\text{FID}_k \downarrow$ | $\text{FID}_g \downarrow$ | $\Delta_{\text{Pose}} \uparrow$ | $\Delta_{\text{Trans}} \uparrow$ | Freezing $\downarrow$ | $\text{Dist}_k \uparrow$ | $\text{Dist}_g \uparrow$ | BeatAlign $\uparrow$ |
| N=32 | 29.74 | 17.34 | 1.53 | 1.32 | 32.9% | 7.50 | 6.49 | 0.245 |
| N=64 | 29.14 | 16.98 | 1.54 | 1.33 | 31.7% | 7.56 | 6.52 | 0.246 |
| N=128 | 27.31 | 14.86 | 1.61 | 1.35 | 30.2% | 7.62 | 6.54 | 0.248 |
| N=256 | **25.96** | **13.42** | **1.64** | **1.36** | **29.6%** | **7.68** | **6.59** | **0.249** |
| N=512 | 26.39 | 14.07 | 1.63 | 1.35 | 29.9% | 7.64 | 6.56 | 0.248 |

Table 6: Evaluation of our module applicability to other models for dance generation.

| Method | Quality | | Diversity | |
|---|---|---|---|---|
| | $\text{FID}_k \downarrow$ | $\text{FID}_g \downarrow$ | $\text{Dist}_k \uparrow$ | $\text{Dist}_g \uparrow$ |
| Revolution [11] | 73.42 | 25.92 | 3.52 | 4.87 |
| + Our Modules | **50.83** | **23.75** | **5.13** | **5.69** |

Table 7: Evaluation of our module applicability to other models for motion prediction.

| Method | Reconstruction error $\downarrow$ | | | | |
|---|---|---|---|---|---|
| | 10 | 20 | 30 | 40 | 50 |
| PHD [40] | 64.4 | 67.1 | 81.1 | 98.5 | 125.9 |
| + Our Modules | **62.9** | **64.3** | **77.2** | **93.5** | **117.8** |

We visualize the poses in the generated dances using t-SNE [34] in Figure 6. We can see that the dances generated by the baseline tend to be mixed together with other genres. It means the motions may have lower fidelity, smaller magnitude and lack uniqueness. We think this is caused by the freezing problem and the high spatio-temporal complexity of the prediction space. In contrast, adding RefineBank alleviates the freezing problem which allows the poses to preserve the high-fidelity details and to be differentiable from other dances. Further adding TransitBank allows to generate more diverse dances with larger motion magnitude.

**Bank-based Auto-encoder** We compare our bank-based auto-encoder with other options such as vanilla AE and VAE. The experimental results are shown in Table 3. Our bank-based auto-encoder achieves clearly better results than the other methods. This is because the bank of latent codes provide a tight characterization of the dance manifold. The tightness requires that the generated motion sequences strictly follow the dance styles.

We also compare to a discrete variant of Bank-AE. Different from our current method, it uses the closest bank item instead of convex combinations of the neighboring items to reconstruct each datum similar to VQ-VAE [33]. We can see that it also reduces the the freezing rate. However, the quality and diversity metrics are notably worse than our method. This is because the discrete variant has

limited capability to reconstruct data with sufficient accuracy, compared to our approach using convex combinations of multiple bank items.

**Number of Latent Features**    We study the impact of the number of latent features $n$ for representing a motion segment as discussed in section 4.1. The results are shown in Table 4. In general, using more latent features improves the prediction performance because it can capture more details. But keep increasing the number may lead to degeneration.

**Number of Bank Items**    We study the impact of the number of elements $N$ in the two banks. The results are shown in Table 5. Initially, increasing the number of elements improves the results. This is reasonable because the expressive power of the bank is improved and more details can be preserved after projection. However, keep increasing $N$ to $512$ begins to have negative effects. We think this is because using too many items may increase the risk of over-fitting to the small levels of noise in the GT data. In addition, the introduced redundancy may also bring negative effects. Nevertheless, the method is relatively robust to this parameter and achieves reasonably good results when $N$ is between $128$ and $512$.

**Evaluation of Applicability**    Our proposed two modules RefineBank & TransitBank can be plugged into other prediction-based methods to achieve long-term non-freezing dance generation. As shown in table 6, our proposed modules can clearly improve motion quality and generation diversity over the RNN-based prediction method Dance Revolution [11].

Besides the adaption of other approaches for dance generation, we also try to adapt our proposed modules to the more general 3D human motion prediction task. We add our proposed modules to a strong open source method PHD [40]. Following the settings in [40], we report PA-MPJPE (mean error of per joint position after applying Procrustes Alignment [7]) in mm through 50 frames and evaluate the performance on a large 3D human motion dataset Human3.6M [13]. As shown in Table 7, we can observe that our proposed approach can clearly improve the prediction accuracy, which demonstrates that our proposed approach can be used for other tasks related to human motion.

# 6    Conclusion

In this work, we presented two general modules that can be plugged into the existing methods to address the freezing problem in dance motion generation. This largely overlooked problem has limited motion generation to short segments of only several seconds. By reducing noise accumulation and exploiting dynamic priors, our approach can generate motions for at least 30 seconds with 60 FPS, which is the maximum music length in current dataset AIST++, without freezing. Our user study also shows that our method has obvious advantages over other methods in terms of quality and diversity. The method paves the way for addressing a more valuable problem of full choreography for entire musics instead of short clips.

**Broader Impact**    We believe that our work has values for not only dance generation but also for more general motion prediction. This benefits areas including media platforms, robotics, and autonomous driving. On the other hand, our method can have negative downstream consequences such as being extended to generate fake videos conditioned on the generated human motions with GANs. The potential limitation is that due to the relatively short duration of music pieces in AIST++ dataset, the test performance is not a precise evaluation for long-term dance generation capability.

## Acknowledgments and Disclosure of Funding

Jian-Fang Hu is the corresponding author. This work was supported partially by the NSFC (U1911401, U1811461, 62076260, 61772570), Guangdong Natural Science Funds Project (2020B1515120085), Guangdong NSF for Distinguished Young Scholar (2022B1515020009), and the Key-Area Research and Development Program of Guangzhou (202007030004). We thank Wenjun Zeng from Eastern Institute for Advanced Study and Dewen Ai from An-Ji Mixed Reality Research Institute for helpful discussions. We also thank Ruilong Li and Siyao Li for sharing their generation results.

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
