# OpenReview forum: "You Never Stop Dancing: Non-freezing Dance Generation via Bank-constrained Manifold Projection"
_NeurIPS.cc/2022/Conference — NeurIPS 2022 Accept_

### Official Review · Reviewer_1dJr · 2022-07-07

**Rating:** 6
**Confidence:** 4
**Soundness:** 3 good
**Presentation:** 4 excellent
**Contribution:** 2 fair

**Summary:**

A common failure mode of human motion or dance generation models (which aim to synthesize parametrized human pose sequences, possibly conditioned on music), is that when attempting to predict long sequences of motion, they can freeze. Essentially, due to accumulated prediction uncertainty, the model eventually decides to predict a static, non-moving pose, as a way to minimize prediction error. This has been raised in a number of papers, and there are a few ideas on how to solve it, but no clear definition or metric exists.

The authors address this by adapting the method, dataset, and evaluation from the FACT [18] paper, and adding two modules designed to eliminate freezing, as well as proposing a new freezing metric.

The modules, RefineBank, and TransitBank, are based on attention-style lookups from learned memory banks. Specifically, RefineBank is designed to take a predicted motion segment, and clean it up by projecting it to and from a low-dimensional motion manifold. Essentially use a transformer encoder to the pose input sequence, do memory lookup for each pose from this learned pose bank, then decode the sequence. TransitBank uses a similar technique to propose extensions of an initial pose sequence based on a learned memory bank that maps motion prefixes to suffixes.

The new freezing metric is based on how often the pose delta or global translation delta between frames is less than a threshold determined by analyzing the ground truth data. The proposed model improves upon FACT in standard FID, Diversity, and beat alignment methods, while doing significantly better at presenting freezing, and is preferred in a human quality assessment study.



**Questions:**

Table 1 copies the numbers from [18] Table 2, but misreports two numbers: for FACT, FIDg is 12.40 and Distg is 5.30. Using the numbers from [18], "Ours" would no longer surpass the original FACT on FIDg. Could the authors please explain?

How does the proposed method for presenting freezing interact with FACT's "future-n full-attention" scheme, also designed to prevent freezing? Do RefineBank & TransitBank eliminate the need for future-n full-attention?


**Limitations:**

yes

**Strengths And Weaknesses:**

**Quality & Clarity:** The paper is clearly written and easy to follow. In particular, the diagrams in Figures 1-4 were well done, and very helpful for understanding sections 4.1-4.3. The explanations are sound, with two exceptions detailed below.

Line 84: SPML should be cited and explained. Also the phrase "Since shapes" should be explained more clearly, to convey that it's the shape parameters of the individual dancers you're discarding.

Line 182: "We construct a separate manifold bank and past-future bank for each dance genre." This implies that, at test time, the model is informed which dance genre should be used for generation. This is strictly more information than is used in FACT [18], potentially making the problem easier. The need for per-genre memory banks and the quantitative advantage obtained by doing so should be addressed further.

Lines 210-213 "Interestingly, the ∆ Trans of our method is even larger than that of GT. ..." This explanation highlights a problem with the definition of the delta trans, delta pose, and Freeze metrics. The goal shouldn't be to maximize delta pose & delta trans, and minimize Freeze... instead it should be for the predicted model to hit the same values for each sequence as observed in the ground truth.

Furthermore, I cannot square the claim that "it is difficult for learning-based methods to predict stationary poses due to the lack of sufficient data" with the observation that the "GT often has stationary poses at transition moments". This suggests that there *is* sufficient stationary data, and the models are not capturing it, or specifically designed to never freeze even though occasional freezes are desirable.

**Originality & Significance:** This paper is an extension of the FACT model from [18] with a small enhancement based on relatively known techniques (transformer memory) and new evaluation metric, applied to the same AIST++ dataset. As such the originality and significance is very minor. Perhaps if the memory bank methods were demonstrated to be more widely useful across a variety of human motion tasks or models, the paper would be more significant. As is, this does not meet the bar for publication based on significance and originality.

---

> ### Author Response · Authors · 2022-08-02
> **Response To Reviewer 1dJr (First Part)**
>
> Thanks to the reviewer for the constructive comments. We have carefully addressed your concerns and provided detailed responses for each review.
>
> **Q1. This paper is an extension of the FACT model from [18] with a small enhancement based on relatively known techniques (transformer memory) and new evaluation metric, applied to the same AIST++ dataset. As such the originality and significance is very minor.**
>
> Re: We would like to point out that our work is not simply an extension of the FACT model. First, our key contribution lies in finding that error accumulation should be responsible for the freezing problem in auto-regressive models. This has values for long-term motion prediction. Second, we present a concrete implementation (i.e., RefineBank) to reduce the noises in the predicted motions, which in turn alleviates the freezing problem. Besides, we propose TransitBank to model the coherence in <past, future> motion dynamics, which can help reduce the uncertainty and ambiguity in motion prediction and effectively facilitate higher-fidelity and diverse motion generation.
>
> Actually, our proposed two modules RefineBank & TransitBank can be plugged into most motion-prediction-based methods to achieve long-term non-freezing dance generation. Since FACT is the previous state-of-the-art work, we conduct experiments on this strong baseline to demonstrate the effectiveness of our proposed modules. To verify the applicability, we add our proposed modules to another method Dance Revolution [10] and detailed results are presented in the following table. Our proposed modules can clearly improve motion quality, generation diversity and music-dance alignment over this RNN-based prediction method.
>
> |                                   |         |         | Quality |          |            | Diversity |          | Alignment   |
> | --------------------------------- | ------- | ------- | ------- | -------- | ---------- | --------- | -------- | ----------- |
> |                                   | FID_k ↓ | FID_g ↓ | ∆Pose ↑ | ∆Trans ↑ | Freezing ↓ | Dist_k ↑  | Dist_g ↑ | BeatAlign ↑ |
> | Dance Revolution                  | 73.42   | 31.01   | -       | -        | -          | 3.52      | 2.46     | 0.220       |
> | Dance Revolution With Our Modules | 50.83   | 23.75   | -       | -        | -          | 5.13      | 5.69     | 0.235       |
>
> **Q2. Perhaps if the memory bank methods were demonstrated to be more widely useful across a variety of human motion tasks or models, the paper would be more significant.**
>
> Re: Thank you for the suggestion. Besides the adaption of another approach for dance generation, we try to adapt our proposed modules to the more general 3D human motion prediction task. The goal of 3D human motion prediction is to predict future 3D human motion from past RGB frames. We add our proposed modules to a strong open source method PHD [R.1]. Following the settings in [R.1], we report PA-MPJPE (mean error of per joint position after applying Procrustes Alignment [R.2]) in mm and evaluate the performance on a large 3D human motion dataset Human 3.6M [R.3] with 15 indoor activities such as discussion, walking and taking photos. Similar to the implementation for dance generation task, we also construct Manifold Bank and Past-Future Bank for each action class. We generate human motion through 50 frames and also report our proposed range-related metrics. Detailed results are presented in the following table. We can observe that our proposed approach can both improve the prediction accuracy and enhance the motion range, which demonstrates that our proposed approach can be used for other tasks related to human motion.
>
> |                      |      |      | PA-MPJPE ↓ |      |       |         | Range    |            |
> | -------------------- | ---- | ---- | ---------- | ---- | ----- | ------- | -------- | ---------- |
> |                      | 10   | 20   | 30         | 40   | 50    | ∆Pose ↑ | ∆Trans ↑ | Freezing ↓ |
> | PHD                  | 64.4 | 67.1 | 81.1       | 98.5 | 125.9 | 0.37    | 0.23     | 28.3%      |
> | PHD With Our Modules | 62.9 | 64.3 | 77.2       | 93.5 | 117.8 | 0.46    | 0.28     | 22.8%      |
>
> **Q3. Line 84: SPML should be cited and explained. Also the phrase "Since shapes" should be explained more clearly, to convey that it's the shape parameters of the individual dancers you're discarding.**
>
> Re: Thank you for the suggestion. We will cite SPML [R.4] in the revision. The shape parameter in SMPL is a vector of 10 scalar values, each of which could be interpreted as an amount of expansion/shrink of a human subject along some direction such as taller or shorter. Since this is irrelevant to motions, we discard the shape parameters of individual dancers.

---

> ### Author Response · Authors · 2022-08-02
> **Response To Reviewer 1dJr (Second Part)**
>
> **Q4. Line 182: "We construct a separate manifold bank and past-future bank for each dance genre." This implies that, at test time, the model is informed which dance genre should be used for generation. This is strictly more information than is used in FACT [18], potentially making the problem easier. The need for per-genre memory banks and the quantitative advantage obtained by doing so should be addressed further.**
>
> Re: In practice, when professionals create a dance for a piece of music, they already have something in mind about the dance genre. So it seems natural to leverage such genre information in choreography to achieve consistency in styles. In addition, considering that the number of genres is not large in practice, it doesn't have scalability issues. Nevertheless, we conducted the experiment of learning a single model for all genres (termed as Genre-agnostic Bank) and showed the results in the Table below. Our approach using genre-agnostic bank also brings notable improvement over the baseline.
>
> |                     |         |         | Quality |          |            | Diversity |          | Alignment   |
> | ------------------- | ------- | ------- | ------- | -------- | ---------- | --------- | -------- | ----------- |
> |                     | FID_k ↓ | FID_g ↓ | ∆Pose ↑ | ∆Trans ↑ | Freezing ↓ | Dist_k ↑  | Dist_g ↑ | BeatAlign ↑ |
> | Baseline            | 35.35   | 22.11   | 1.33    | 1.07     | 39.0%      | 5.94      | 6.18     | 0.241       |
> | Genre-agnostic Bank | 28.57   | 15.92   | 1.58    | 1.33     | 31.9%      | 7.29      | 6.42     | 0.247       |
> | Genre-specific Bank | 25.96   | 13.42   | 1.64    | 1.36     | 29.6%      | 7.68      | 6.59     | 0.249       |
>
> **Q5. Lines 210-213 "Interestingly, the ∆ Trans of our method is even larger than that of GT. ..." This explanation highlights a problem with the definition of the delta trans, delta pose, and Freeze metrics. The goal shouldn't be to maximize delta pose & delta trans, and minimize Freeze... instead it should be for the predicted model to hit the same values for each sequence as observed in the ground truth.**
>
> Re: We agree that ∆Trans alone is not perfect. In particular, larger ∆Trans doesn't mean better dance quality. We should jointly consider different metrics to get a more comprehensive understanding of the predicted motions. The reason why we still want to discuss it in the main paper is to show the limitation of our method, i.e., it doesn't generate appropriate stationary poses at transition moments.
>
> **Q6. I cannot square the claim that "it is difficult for learning-based methods to predict stationary poses due to the lack of sufficient data" with the observation that the "GT often has stationary poses at transition moments". This suggests that there is sufficient stationary data, and the models are not capturing it, or specifically designed to never freeze even though occasional freezes are desirable.**
>
> Re: Thank you for the suggestion. It is very helpful to truly understand the problem. While GT often has stationary poses at transition moments, they only occupy a small portion of all poses. As a result, most methods, including ours and the baseline, cannot handle it well. There are many works on learning from unbalanced data which may be helpful to address the problem. We leave it for our future work.
>
> **Q7. Table 1 copies the numbers from [18] Table 2, but misreports two numbers: for FACT, FIDg is 12.40 and Distg is 5.30. Using the numbers from [18], "Ours" would no longer surpass the original FACT on FIDg. Could the authors please explain?**
>
> Re: Authors of FACT [18] said there is some mistake in their geometric feature computation and they updated the correct version on the github. Thus we recompute metrics based on geometric feature (e.g., FID_g & Dist_g) using the officially updated evaluation code.

---

> ### Author Response · Authors · 2022-08-02
> **Response To Reviewer 1dJr (Third Part)**
>
> **Q8. How does the proposed method for presenting freezing interact with FACT's "future-n full-attention" scheme, also designed to prevent freezing? Do RefineBank & TransitBank eliminate the need for future-n full-attention?**
>
> Re: Thank you for the suggestion. We implemented our model and the baseline with both shift-by-1 casual-attention and Future-n Full-attention, respectively. The detailed results are presented in the table below. We can see that both replacing Shift-by-1 Causal-attention with Future-n Full-attention and adding our bank solution indeed reduce the freezing rate. And combining these two designs will further reduce the freezing rate notably.
>
> |                                        |         |         | Quality |          |            | Diversity |          | Alignment   |
> | -------------------------------------- | ------- | ------- | ------- | -------- | ---------- | --------- | -------- | ----------- |
> |                                        | FID_k ↓ | FID_g ↓ | ∆Pose ↑ | ∆Trans ↑ | Freezing ↓ | Dist_k ↑  | Dist_g ↑ | BeatAlign ↑ |
> | Baseline (Shift-by-1 Casual-attention) | 111.69  | 34.59   | 0.95    | 0.73     | 52.4%      | 4.13      | 4.65     | 0.217       |
> | Ours (Shift-by-1 Casual-attention)     | 41.91   | 25.32   | 1.37    | 1.12     | 37.4%      | 6.07      | 6.24     | 0.237       |
> | Baseline (Future-n Full-attention)     | 35.35   | 22.11   | 1.33    | 1.07     | 39.0%      | 5.94      | 6.18     | 0.241       |
> | Ours (Future-n Full-attention)         | 25.96   | 13.42   | 1.64    | 1.36     | 29.6%      | 7.68      | 6.59     | 0.249       |
>
> Reference:
>
> [R.1] Zhang, Jason Y., et al. "Predicting 3d human dynamics from video." *ICCV*. 2019.
>
> [R.2] Gower, John C. "Generalized procrustes analysis." *Psychometrika* 40.1 (1975): 33-51.
>
> [R.3] Ionescu, Catalin, et al. "Human3. 6m: Large scale datasets and predictive methods for 3d human sensing in natural environments." *IEEE transactions on pattern analysis and machine intelligence* 36.7 (2013), 1325-1339
>
> [R.4] Loper, Matthew, et al. "SMPL: A skinned multi-person linear model." *ACM transactions on graphics (TOG)* 34.6 (2015): 1-16.

---

### Official Review · Reviewer_HLRW · 2022-07-07

**Rating:** 7
**Confidence:** 4
**Soundness:** 3 good
**Presentation:** 3 good
**Contribution:** 2 fair

**Summary:**

The authors look into the problem of freezing motions in dance generation frameworks.  They propose RefineBank and TransitBank modules that can be plugged on existing methods to generate high fidelity dance movements without freezing after a few frames. Refine Bank learns a low dimensional manifold representaton of motion data close to the ground truth to prevent error accumulation. TransitBank learns a motion prior for the future poses to reduce ambiguity in long-term motion prediction.The authors evaluated their method on the AIST++ dataset and outperform certain baselines. The authors also propose a new metric which measues rate of freezing sub-sequences in movements.

**Questions:**

- How is the commitment loss different from the item loss (Eq. 7)?
- How is the Win Rate interpreted in the last column of Table 1?

**Limitations:**

- Since this is an incremental work on the existing architecture of FACT [18], it is heavily reliant on the performance of the baseline FACT model. It is unclear if or how the proposed modules can be combined with other baseline models such as [10, 17, 38].

- It is also unclear if the proposed modules can be generalized to any genre of dancing.

- Qualitative results also produce some self-intersections of the limbs, which is problematic for such synthesis problems.

**Strengths And Weaknesses:**

Strengths:
- The authors tackle a challenging problem for long-term dance motion generation and synthesize diverse and natural dance movements with respect to music. The presented qualitative results and user study results indicate the proposed approach is promising.
- Using the bank-based auto-encoder modules in the architecture to learn a compact motion manifold is technically sound.
- The authors provide ablations for various experiments to strengthen their claims.
- The paper is overall clearly written and easy to follow.

Weaknesses:
- I am not clear why the three-stage optimization is required. Why is the Manifold Bank trained separately from the Refine Bank encoder and decoder?
- How are the different loss terms being used in the first stage of optimization (Eq. 7)? Ablations for the use of multiple losses are missing.

---

> ### Author Response · Authors · 2022-08-02
> **Response To Reviewer HLRW (First Part)**
>
> Thanks to the reviewer for the constructive comments. We have carefully addressed your concerns and provided detailed responses for each review.
>
> **Q1. I am not clear why the three-stage optimization is required. Why is the Manifold Bank trained separately from the Refine Bank encoder and decoder?**
>
> Re: The main reason for adopting stage-wise training is to ensure that the bank (learned in stage one) can accurately reconstruct the "GT motions". In contrast, if we adopt end-to-end training, the bank elements will be learned from the predicted noisy motions. During the rebuttal period, we tried the end-to-end training strategy and showed the detailed results in the table below. We can see that the stage-wise training strategy has clear advantages in terms of all metrics.
>
> |                             |         |         | Quality |          |            | Diversity |          | Alignment   |
> | --------------------------- | ------- | ------- | ------- | -------- | ---------- | --------- | -------- | ----------- |
> |                             | FID_k ↓ | FID_g ↓ | ∆Pose ↑ | ∆Trans ↑ | Freezing ↓ | Dist_k ↑  | Dist_g ↑ | BeatAlign ↑ |
> | Without Stage-wise Training | 29.37   | 16.75   | 1.52    | 1.29     | 33.4%      | 6.73      | 6.39     | 0.246       |
> | With Stage-wise Training    | 25.96   | 13.42   | 1.64    | 1.36     | 29.6%      | 7.68      | 6.59     | 0.249       |
>
> **Q2. How are the different loss terms being used in the first stage of optimization (Eq. 7)? Ablations for the use of multiple losses are missing.**
>
> Re: Actually, inspired by VQ-VAE [29], we design a similar three-part loss function to train our Manifold Bank in the first training stage. The first term minimizes the reconstruction error to ensure the quality of reconstructed motion. The second term is the item loss to update items in the bank. This objective function moves the items close to the outputs of the encoder. Thus we can use one item in the Manifold Bank to construct a motion sequence. The third term is the commitment loss to make sure the output of encoder commits to an embedding. Thus we can ensure that the output of encoder does not grow arbitrarily and the two spaces (i.e., encoding space and embedding space) are well aligned. We also conduct an ablation study to train Manifold Bank with only the first two loss terms and detailed results are presented in the following table. As shown, introducing commitment loss can clearly improve motion quality but only marginally reduce freezing rate. One possible reason is that introducing manifold projection can effectively prevent error accumulation and the alignment of these two spaces will affect the reconstruction quality.
>
> |                         |         |         | Quality |          |            | Diversity |          | Alignment   |
> | ----------------------- | ------- | ------- | ------- | -------- | ---------- | --------- | -------- | ----------- |
> |                         | FID_k ↓ | FID_g ↓ | ∆Pose ↑ | ∆Trans ↑ | Freezing ↓ | Dist_k ↑  | Dist_g ↑ | BeatAlign ↑ |
> | Without Commitment Loss | 28.03   | 15.31   | 1.60    | 1.34     | 30.5%      | 7.34      | 6.47     | 0.248       |
> | With Commitment Loss    | 25.96   | 13.42   | 1.64    | 1.36     | 29.6%      | 7.68      | 6.59     | 0.249       |
>
> **Q3. How is the commitment loss different from the item loss (Eq. 7)?**
>
> Re: The item loss is designed to update items in the bank. This objective function moves the items close to the outputs of the encoder. Thus we can use one item in the Manifold Bank to construct a motion sequence. The commitment loss is designed to make sure the output of encoder commits to an embedding. Thus we can ensure that the output of encoder does not grow arbitrarily and the two spaces (i.e., encoding space and embedding space) are well aligned.
>
> **Q4. How is the Win Rate interpreted in the last column of Table 1?**
>
> Re: As shown in subsection User Study (L219-L230), We invite 20 participants and each participant is asked to watch 30 pairs of comparison videos. Each pair consists of our dance and one competitor’s generated with the same music. We ask each participant to determine “which person is dancing better to the music” and compute the win rate of our method. Taking FACT as an example,  87% of our generated dance motion is considered better than FACT.

---

> ### Author Response · Authors · 2022-08-02
> **Response To Reviewer HLRW (Second Part)**
>
> **Q5. Since this is an incremental work on the existing architecture of FACT [18], it is heavily reliant on the performance of the baseline FACT model. It is unclear if or how the proposed modules can be combined with other baseline models such as [10, 17, 38].**
>
> Re: We would like to point out that our work is not simply an incremental work on the existing architecture of FACT. First, our key contribution lies in finding that error accumulation should be responsible for the freezing problem in auto-regressive models. This has values for long-term motion prediction. Second, we present a concrete implementation (i.e., RefineBank) to reduce the noises in the predicted motions, which in turn alleviates the freezing problem. Besides, we propose TransitBank to model the coherence in <past, future> motion dynamics, which can help reduce the uncertainty and ambiguity in motion prediction and effectively facilitate higher-fidelity and diverse motion generation.
>
> Actually, our proposed two modules RefineBank & TransitBank can be plugged into most motion-prediction-based methods to achieve long-term non-freezing dance generation. Since FACT is the previous state-of-the-art work, we conduct experiments on this strong baseline to demonstrate the effectiveness of our proposed modules. To verify the applicability, we add our proposed modules to another method Dance Revolution [10] and detailed results are presented in the following table. Our proposed modules can clearly improve motion quality, generation diversity and music-dance alignment over this RNN-based prediction method. It is worth mentioning that our proposed approach can be used for other tasks (e.g., 3D human motion prediction) related to human motion. Detailed analysis about such applicability can be found in our response to Q2 for Reviewer 1dJr.
>
> |                                   |         |         | Quality |          |            | Diversity |          | Alignment   |
> | --------------------------------- | ------- | ------- | ------- | -------- | ---------- | --------- | -------- | ----------- |
> |                                   | FID_k ↓ | FID_g ↓ | ∆Pose ↑ | ∆Trans ↑ | Freezing ↓ | Dist_k ↑  | Dist_g ↑ | BeatAlign ↑ |
> | Dance Revolution                  | 73.42   | 31.01   | -       | -        | -          | 3.52      | 2.46     | 0.220       |
> | Dance Revolution With Our Modules | 50.83   | 23.75   | -       | -        | -          | 5.13      | 5.69     | 0.235       |
>
> **Q6. It is also unclear if the proposed modules can be generalized to any genre of dancing.**
>
> Re: For dance genres that appear in the training set, we can generate long-term non-freezing dances with the corresponding banks (termed as Genre-specific Bank). For genres not appear in the training set, we can also learn a single model for all genres (termed as Genre-agnostic Bank) and we present detailed results in the Table below. Our approach using genre-agnostic bank can achieve barely satisfactory performance and also brings notable improvement over the baseline.
>
> |                     |         |         | Quality |          |            | Diversity |          | Alignment   |
> | ------------------- | ------- | ------- | ------- | -------- | ---------- | --------- | -------- | ----------- |
> |                     | FID_k ↓ | FID_g ↓ | ∆Pose ↑ | ∆Trans ↑ | Freezing ↓ | Dist_k ↑  | Dist_g ↑ | BeatAlign ↑ |
> | Baseline            | 35.35   | 22.11   | 1.33    | 1.07     | 39.0%      | 5.94      | 6.18     | 0.241       |
> | Genre-agnostic Bank | 28.57   | 15.92   | 1.58    | 1.33     | 31.9%      | 7.29      | 6.42     | 0.247       |
> | Genre-specific Bank | 25.96   | 13.42   | 1.64    | 1.36     | 29.6%      | 7.68      | 6.59     | 0.249       |
>
> **Q7. Qualitative results also produce some self-intersections of the limbs, which is problematic for such synthesis problems.**
>
> Re: We also find this phenomenon and extra discriminators will reduce the self-intersection. Considering that this problem has been well explored in human motion reconstruction, we do not focus on it in this work and leave it as future work.

---

> ### Comment · Reviewer_HLRW · 2022-08-07
> **Thanks for the detailed response**
>
> I thank the authors for their detailed response, which addresses my main concerns. The paper presents a technically sound approach to solving a challenging problem, which is well-supported by the experimental results. For these reasons, I maintain my vote for acceptance. I would recommend the authors add qualitative results of the genre-agnostic bank to their supplementary results (i.e., performance on genres that do not appear in training) to highlight the generalizability of their approach

---

### Official Review · Reviewer_53W8 · 2022-07-08

**Rating:** 6
**Confidence:** 4
**Soundness:** 2 fair
**Presentation:** 1 poor
**Contribution:** 2 fair

**Summary:**

This submission specifically focuses on preventing the generation of frozen motions in long-term motion generation task.
To achieve this goal, the latent code (z^hat) used to generate the final motion is represented as the linear combination of codes in a learned codebook (termed as feature bank in this submission) as VQ-VAE [29]. And the weight of every code (in the codebook) is determined by the similarity between the original latent code (z) and the code.
And a TransitBank is proposed to "retrive" future motion features according to the past motion features.

**Questions:**

The novelty; More thorough evaluation and analysis; Much improved writing.

**Strengths And Weaknesses:**

**Major weaknesses**

- The description is not clear enough and I could not understand the exact operations of the proposed method.

L94, L156: cite the VQ-VAE [29] paper. What is the difference between the proposed RefineBank and VQ-VAE? This is very important and should be discussed as thorough as possible. Maybe you use the weighted sum of the codes while they use a certain code? Personally, I do not think this is a good choice. Pros and cons of these two choices should be evaluated. BTW, why do you need to do a clustering in L143-144? And what is the E_M, is it the same as E_RB?

Since there is an average operation when calculating b^i (L145), so I guess the network extract a latent code for every frame (but with a temporal receptive field), right? Then what are the Ks/Vs in the past-future bank, I did not find the T dimension but it's said that "TransitBank is a temporal extension of RefineBank" in L139? Frankly, I did understand the exact use and learning related to TransitBank (Sec. 4.2 & 4.3). Many operations need to be clarified.

BTW, try not to name both network module (RefineBank) and tensor (manifold bank) as banks. It is a little confusing. Personally, I would recommend using code (codebook) or word (vocabulary) to refer to the latent feature (the collection of the latent features).

How does the music feature (Fig. 1) affect the final output? Is it working? Some dedicated experiments should be conducted about this part.

Why does FACT results look that bad even for the first few seconds?

L119: "basic short motion segments": What are they and how do you find them?

User study: Are there any professionals conducting the user study (L220)? Better in what aspect (L222)? Normally, people will compare several specific and clearly described aspects in a user study.

**Minor weaknesses**

L175: How many sec is 240 frames of music?

L197: differences between what and what?

L212: What do you mean by "stationary poses at transition moments"?

Fig. 6:How is GT looks like?

What are prediction-based methods (L57), retrieval-based methods (L68), and bank-based auto-encoder (L249) exactly? These are not well-known concept yet.

---

> ### Author Response · Authors · 2022-08-02
> **Response To Reviewer 53W8 (First Part)**
>
> Thanks to the reviewer for the constructive comments. We have carefully addressed your concerns and provided detailed responses for each review.
>
> **Q1: Novelty**
>
> Re: First, we would like to point out that our key contribution lies in finding that error accumulation should be responsible for the freezing problem in auto-regressive models. This has values for long-term motion prediction. Second, we present a concrete implementation (i.e., RefineBank) to reduce the noises in the predicted motions, which in turn alleviates the freezing problem. Besides, we propose TransitBank to model the coherence in <past, future> motion dynamics, which can help reduce the uncertainty and ambiguity in motion prediction and effectively facilitate higher-fidelity and diverse motion generation. It is worth mentioning that our proposed two modules RefineBank & TransitBank can be plugged into most motion-prediction-based methods to achieve long-term non-freezing dance generation. Detailed analysis about applicability can be found in our response to Q1&Q2 for Reviewer 1dJr.
>
> **Q2. The major weakness is that the description is not clear enough and I could not understand the exact operations of the proposed method.**
>
> Re: We provide detailed clarifications of the raised questions in the following, hoping to help you understand the work better.
>
> **Q3. L94, L156: cite the VQ-VAE [29] paper. What is the difference between the proposed RefineBank and VQ-VAE? This is very important and should be discussed as thorough as possible. Maybe you use the weighted sum of the codes while they use a certain code? Personally, I do not think this is a good choice. Pros and cons of these two choices should be evaluated.**
>
> Re: We would like to point out that our core contribution is to systematically show that error accumulation in auto-regressive models should be responsible for the freezing problem in long-term motion prediction, and present a solution (i.e., bank-constrained manifold projection) based on manifold learning to remove the noises in the predicted motions. The proposed manifold projection can be implemented by our designed RefineBank or other bank-based autoencoders such as VQ-VAE, and thus VQ-VAE is just one alternative way to implement the idea (termed as Bank-AE (Discrete)). We adopt a slightly different variant (termed as Bank-AE (Ours)) in which we use soft-assignment instead of hard nearest-neighbor assignment. This allows us to reconstruct motions with higher precision. As shown in Table 3 in the main paper, both implementations can improve generation performance and introducing soft-assignment is very important in improving both the quality and diversity of generated dances. It is worth mentioning that our proposed solution can also work well in combination with other manifold learning methods such as vanilla AE and VAE.
>
> **Q4. BTW, why do you need to do a clustering in L143-144?**
>
> Re: As shown in L142-146, we regard the cluster centers as potential basic motion segments and train Manifold Bank to reconstruct motion sequences of these cluster centers. Thus we can obtain a proper initialization of Manifold Bank where items are representative and widely used in GT motion. This operation is only to provide a better initialization. Without clustering, the convergence speed will become slower.
>
> **Q5. What is the E_M, is it the same as E_RB?**
>
> Re: The architecture of E_M is the same as E_RB. The learned parameters of E_M during the first training stage can also be used as the initialization for E_RB. Then the parameters of E_RB will be updated during the second training stage with reconstruction loss represented in Equation 8.
>
> **Q6. Since there is an average operation when calculating b^i (L145), so I guess the network extract a latent code for every frame (but with a temporal receptive field), right? Then what are the Ks/Vs in the past-future bank, I did not find the T dimension but it's said that "TransitBank is a temporal extension of RefineBank" in L139?**
>
> Re: We apologize that the phrase “temporal extension” may be a little confusing. What we wanted to say is that TransitBank maps motion prefixes to suffixes. As shown in Figure 3, TransitBank is constructed to handle explicit priors about past-future relation on top of the manifold by querying past-future latent feature pairs.
>
> **Q7. BTW, try not to name both network module (RefineBank) and tensor (manifold bank) as banks. It is a little confusing. Personally, I would recommend using code (codebook) or word (vocabulary) to refer to the latent feature (the collection of the latent features).**
>
> Re: Thank you for the suggestion. We will consider replacing the latent feature (the collection of the latent features) with code (codebook), which may help readers understand our work intuitively.

---

> ### Author Response · Authors · 2022-08-02
> **Response To Reviewer 53W8 (Second Part)**
>
> **Q8. How does the music feature (Fig. 1) affect the final output? Is it working? Some dedicated experiments should be conducted about this part.**
>
> Re: In our implementation, we simply followed the setting in FACT and employed Librosa [20] to extract 35-dim music feature. Different types of music features may lead to slightly different generated dances, but this is not the focus of this work.
>
> **Q9: Why does FACT results look that bad even for the first few seconds?**
>
> Re: In our observation, the dance generated by FACT only looks good for the first two seconds, and then the range of motion becomes smaller and even becomes freezing, which is also mentioned in github issues. The possible reason is that after two seconds, input motions to the model will all come from the previously predicted motions which may contain large noises.
>
> **Q10: L119: "basic short motion segments": What are they and how do you find them?**
>
> Re: We consulted with the professional dancers and learned that the choreography for each dance genre does consist of basic choreographic units (i.e., short motion segments), which is also mentioned in previous work [2,33]. Thus we follow this prior and construct Manifold Bank to store such basic motion segments for each dance genre. To reduce the cost of manual labeling, we employed dictionary learning to encourage the model to automatically discover representative motion segments widely used in GT dances.
>
> **Q11: User study: Are there any professionals conducting the user study (L220)? Better in what aspect (L222)? Normally, people will compare several specific and clearly described aspects in a user study.**
>
> Re: Following the implementation in FACT, we invite participants ranging from professional dancers to people who rarely dance. We did not explicitly ask participants to judge who danced better to the music from given aspects but we collected detailed feed-backs, which can be found in L226-230. Our generated dance is generally thought to be more diverse and non-freezing compared with other methods.
>
> **Q12: L175: How many sec is 240 frames of music?**
>
> Re: Frames of music and dance are aligned during feature extraction. Thus the music is also at 60FPS and 240 frames are 4 seconds.
>
> **Q13: L197: differences between what and what?**
>
> Re: This sentence means that we compute the average values of temporal differences of the pose parameters among adjacent frames and average values of the temporal differences of the translation parameters among adjacent frames across the whole sequence.
>
> **Q14: L212: What do you mean by "stationary poses at transition moments"?**
>
> Re: We observe that there exist transition moments between two dance movements, and at this point the dancer becomes stationary or performs small movements without global translation.
>
> **Q15: Fig. 6:How is GT looks like?**
>
> Re: We do not present the GT motion since GT motion of this sample only lasts 8 seconds. In brief, the GT motion does not become freezing and repeats the body swaying from side to side.
>
> **Q16: What are prediction-based methods (L57), retrieval-based methods (L68), and bank-based auto-encoder (L249) exactly? These are not well-known concept yet.**
>
> Re: As shown in L25-26 & L57, prediction-based methods treat dance generation as music conditioned human motion prediction. In L68-70, retrieval-based methods compose a complete dance by retrieval. They select the closest predefined motion segments in a pre-built database based on music, and construct a sequence with the proper transition routines. In L36-38, RefineBank equips an auto-encoder with a bank of latent codes to tightly constrain the manifold to be close to the ground-truth (GT) motions and meanwhile far from the ones with noises.

---

> > ### Comment · Reviewer_53W8 · 2022-08-06
> > **Discussion**
> >
> > - about Q1: "...key contribution lies in finding...": Only L27-28 is related to this claim, right? I did not find in-depth analysis/reasoning or experiments to support this claim either. I could not agree this is a contribution yet until I see more direct support such as dedicated and controlled experiments to study this effect. Many previous works reasonably guess out-of-distribution predictions is related to this effect while none of them claimed this as a contribution.
> >
> > - about Q8: Sorry I was not clear enough. I meant to ask if changing the input audio (actually audio features instead of the raw audio wave is used as input) really changes the generated dance motion. If the audio signal really controls the generated output, I do not see why the network can only generate frozen output since the input audio is changing every frame. This is why dedicated experiments on this aspect are important. Maybe, in this task, AR-based/prediction-based methods eventually generate frozen output because the audio input fails to control the motion. Can you also provide non-averaged (across time) beat alignment results?

---

> > ### Comment · Reviewer_53W8 · 2022-08-06
> > **Discussion 2**
> >
> > - Sec. 4.1 needs a discussion paragraph highlighting the differences between the operations used in this work and that of VQ-VAE. This is **very important** for the readers to clearly tell which contributions belong to VQ-VAE. I would recommend moving L155-163 inside Sec. 4.1 RefineBank too since they are the same as VQ-VAE.
> > - Eq. 2: input motion for TB should be t-k:t, right?
> > - L126: Why is B_PF learned? According to L150-154, it is constructed from B_M, which means as long as B_M (and the weights of RefineBank network) is learned B_PF should be determined too.
> >
> > ### about writing
> > - L129: "query-read process" "attention": Why not use terms like "database, key-value, retrieval" since you make an analogy to retrieval-based methods. You can easily highlight the key differences in this way. As far as I can tell, 1) "retrieval" is done in the latent space, which is beneficial since a type of motions belonging to the same mode could be generated with the help of its decoder; 2) soft assignment instead of nearest search is used (I am not sure if this is beneficial and requires ablation studies).
> > - about Q16: I would recommend formally defining prediction-/retrieval-based methods at the beginning of the Related Work section. And compare/relate the proposed method with the literature at the end of each subsection of the Related Work (e.g. move L75-76 to L67; Extend L77-80 with more specific comparisons, you can use terms, e.g. TransitBank, mentioned in Method and point to them so that the readers can easily get your analogies.)
> > - TransitBank is clearly the major contribution. It is an interesting way to "reduce the uncertainty and ambiguity in motion prediction (L124)" and is conceptually similar to the traditional retrieval-based methods. It is also shown effective on general motion prediction tasks (Q2 in Reviewer 1dJr), which is very good. In contrast, RefineBank is not novel conceptually and technically. In the rebuttal (response to my Q3), the author mentioned it could be other networks than VQ-VAE (if true, please provide some results.) It seems that the RefineBank is just for motion denoising/refinement but helps alleviate error accumulation problem (except that it's also part of the TransitBank in this specific implementation).
> > - In my opinion, this submission needs a substantial restructuring and it is not ready for publication yet.

---

> > > ### Author Response · Authors · 2022-08-09
> > > **Response To Reviewer 53W8 (First Part)**
> > >
> > > **Q1. about Q1: "...key contribution lies in finding...": Only L27-28 is related to this claim, right? I did not find in-depth analysis/reasoning or experiments to support this claim either. I could not agree this is a contribution yet until I see more direct support such as dedicated and controlled experiments to study this effect. Many previous works reasonably guess out-of-distribution predictions is related to this effect while none of them claimed this as a contribution.**
> > >
> > > Re: While some previous works have guessed that out-of-distribution input motions may be related to the freezing effect, most of them use ad hoc solutions of training the models by either alternatively feeding the previously predicted (noisy) motions and GT (noisy-free) motions or manually adding random noise into input motions. However, when the noise distribution in testing is different from that in training, they will still fail. In contrast, we directly project the out-of-distribution motions to the learned manifold and then restore the dance motions with minor noise from the learned manifold space during generation. The experimental results show evidence that reducing the noises in the predicted motions can prevent from generating freezing motions. To the best of our knowledge, this has not been studied previously in this domain. We will revise our claim in the revision.
> > >
> > > **Q2. about Q8: Sorry I was not clear enough. I meant to ask if changing the input audio (actually audio features instead of the raw audio wave is used as input) really changes the generated dance motion. If the audio signal really controls the generated output, I do not see why the network can only generate frozen output since the input audio is changing every frame. This is why dedicated experiments on this aspect are important. Maybe, in this task, AR-based/prediction-based methods eventually generate frozen output because the audio input fails to control the motion. Can you also provide non-averaged (across time) beat alignment results?**
> > >
> > > Re: Changing input audio DOES change the generated motions as observed in our experiments. To show that, we compute the average Euclidean distance in the kinetic (denoted as D_k) and geometric (denoted as D_g) feature space among the generated motions conditioned on the same motion seed but different music. As shown in the following table, both our model and FACT can generate different dances conditioned on different music, with ours being more diverse.
> > >
> > > |      | Diversity |       |
> > > | :--- | --------- | ----- |
> > > |      | D_k ↑     | D_g ↑ |
> > > | FACT | 4.67      | 4.78  |
> > > | Ours | 7.21      | 6.19  |
> > >
> > > We try to share our shallow thoughts on why the networks generate frozen motions even though the input audio is changing every frame. The motion prediction network is a non-linear system where the audio features and motion features have complex interactions. In the case where the motion features are out of distribution, the behavior of the network becomes unpredictable as it is NOT trained on data like them at all. So, it is possible to generate freezing motions even though the audio is changing. Per your request, we have updated the supplemental material and provided the non-averaged (across time) beat alignment results as Figure 4 in the revised supplemental material as a supplement to Figure 5 in the main paper.

---

> > > ### Author Response · Authors · 2022-08-09
> > > **Response To Reviewer 53W8 (Second Part)**
> > >
> > > **Q3. Sec. 4.1 needs a discussion paragraph highlighting the differences between the operations used in this work and that of VQ-VAE. This is very important for the readers to clearly tell which contributions belong to VQ-VAE. I would recommend moving L155-163 inside Sec. 4.1 RefineBank too since they are the same as VQ-VAE.**
> > >
> > > Re: As mentioned in our previous response, one of our contributions lies in presenting a solution based on manifold learning to reduce the noises in the predicted motions, which effectively alleviates the freezing problem in long-term motion prediction. To the best of our knowledge, it is the first work to solve this problem by manifold projection in this domain. The proposed manifold projection can be implemented by our designed RefineBank or other methods such as VQ-VAE, AE, and VAE. VQ-VAE is just one concrete approach to achieve this goal, which is termed as Bank-AE (Discrete) in Table 3 in the main paper. It is worth noting that our method achieves better results than VQ-VAE as will be discussed in below.
> > >
> > > Compared to VQ-VAE, we have two unique designs in our RefineBank for the dance generation task. The first is to use soft-assignment (termed as Bank-AE (Ours)) instead of VQ-VAE's hard nearest-neighbor assignment. As shown in Table 3 in the main paper, using soft-assignment helps improve both the quality and diversity of the generated dances. The second is to take several latent features rather than only one to represent the input motion sequence. As shown in Table 4 in the main paper, using more latent features can improve the prediction performance because it can capture more details. The above analysis was discussed in L254-263. Following your suggestion, we will add a paragraph discussing the differences between our method and VQ-VAE, and move L155-163 to section 4.1.
> > >
> > > **Q4. Eq. 2: input motion for TB should be t-k:t, right?**
> > >
> > > Re: As shown in L180-181, the input to TransitBank is a motion sequence with 120 frames.
> > >
> > > **Q5. L126: Why is B_PF learned? According to L150-154, it is constructed from B_M, which means as long as B_M (and the weights of RefineBank network) is learned B_PF should be determined too.**
> > >
> > > Re: We tried to learn B_PF independently in early attempts. But in the final version, B_PF is directly constructed as the <past, future> extension of B_M since B_M can be interpreted as a reasonable dance manifold after the first stage training. We will revise our paper accordingly.
> > >
> > > **Q6. L129: "query-read process" "attention": Why not use terms like "database, key-value, retrieval" since you make an analogy to retrieval-based methods. You can easily highlight the key differences in this way. As far as I can tell, 1) "retrieval" is done in the latent space, which is beneficial since a type of motions belonging to the same mode could be generated with the help of its decoder; 2) soft assignment instead of nearest search is used (I am not sure if this is beneficial and requires ablation studies).**
> > >
> > > Re: Thank you for the suggestion. We will rewrite the query-read process with the suggested terms. For the required ablation, we implement the variant of TransitBank with nearest search and present detailed results in the following table. We can see that introducing soft-assignment can clearly improve both the quality and diversity of the generated dances.
> > >
> > > |                              |         |         | Quality |          |            | Diversity |          | Alignment   |
> > > | :--------------------------- | :------ | :------ | :------ | :------- | :--------- | --------- | -------- | ----------- |
> > > |                              | FID_k ↓ | FID_g ↓ | ∆Pose ↑ | ∆Trans ↑ | Freezing ↓ | Dist_k ↑  | Dist_g ↑ | BeatAlign ↑ |
> > > | TanksitBank (Nearest Search) | 27.13   | 15.08   | 1.58    | 1.27     | 31.2%      | 6.91      | 6.42     | 0.247       |
> > > | TanksitBank (Ours)           | 25.96   | 13.42   | 1.64    | 1.36     | 29.6%      | 7.68      | 6.59     | 0.249       |
> > >
> > > **Q7. about Q16: I would recommend formally defining prediction-/retrieval-based methods at the beginning of the Related Work section. And compare/relate the proposed method with the literature at the end of each subsection of the Related Work (e.g. move L75-76 to L67; Extend L77-80 with more specific comparisons, you can use terms, e.g. TransitBank, mentioned in Method and point to them so that the readers can easily get your analogies.)**
> > >
> > > Re: Thank you for the suggestion. We will improve the presentation of the related work section to make clear analogies, following your suggestion.

---

> > > ### Author Response · Authors · 2022-08-09
> > > **Response To Reviewer 53W8 (Third Part)**
> > >
> > > **Q8. TransitBank is clearly the major contribution. It is an interesting way to "reduce the uncertainty and ambiguity in motion prediction (L124)" and is conceptually similar to the traditional retrieval-based methods. It is also shown effective on general motion prediction tasks (Q2 in Reviewer 1dJr), which is very good. In contrast, RefineBank is not novel conceptually and technically. In the rebuttal (response to my Q3), the author mentioned it could be other networks than VQ-VAE (if true, please provide some results.)**
> > >
> > > Re: Thank you for appreciating our proposed TranksitBank. As to the RefineBank, we would like to point out that our contribution is to present a solution (i.e., bank-constrained manifold projection) based on manifold learning to remove the noises in the predicted motions and effectively alleviates the freezing problem in long-term motion prediction. The proposed manifold projection can be implemented by our designed RefineBank or other bank-based autoencoders such as VQ-VAE, and thus VQ-VAE is just one alternative way to implement the idea. The detailed comparison with VQ-VAE can be found in our response to Q3 and detailed empirical comparison with other methods has been presented in Table 3 in the main paper.
> > >
> > > **Q9. It seems that the RefineBank is just for motion denoising/refinement but helps alleviate error accumulation problem (except that it's also part of the TransitBank in this specific implementation).**
> > >
> > > Re: We still think RefineBank has values since it shows that by reducing the noises in the input motions it can effectively alleviate the freezing problem. The specific design of RefineBank is related to VQ-VAE but has differences as discussed in our response to Q3. We also empirically validate its advantages in L249-263.
> > >
> > > As you mentioned in Q1, many previous works reasonably guess out-of-distribution prediction is related to freezing motion problem. In the dance generation task, this phenomenon is more pronounced since the reasonable dance manifold is just a minor subset of possible human motion. Thus predicted motions with accumulated error can easily become out of distribution and lead to freezing generation. To solve this problem, we propose a direct solution (i.e., manifold projection or RefineBank) to project out-of-distribution motions into the learned reasonable distribution. This process can also be viewed as removing noises in the predicted motions so that the refined motions can return to the normal distribution. The learned predictor can handle the refined predicted motion better and thus reduce the freezing generation.

---

### Official Review · Reviewer_GKyZ · 2022-07-11

**Rating:** 8
**Confidence:** 4
**Soundness:** 3 good
**Presentation:** 4 excellent
**Contribution:** 4 excellent

**Summary:**

The paper proposes two new modules to improve the motion generation capabilities of auto-regressive networks. The designing of modules lies in the idea of using a bank of learned features, which are used to reduce the noise in prediction motions. Both modules were tested in the task of dance motion generation. The experimental results in the AIST++ dataset show the benefits of using the proposed modules. It is worth mentioning that the modules seem to be very useful in other applications where motion generation is required.

**Questions:**

1. Have you tried to train the network without applying the stage-wise strategy? What would be the performance of the methods?
2. For the sake of reproducibility, kindly consider explicitly saying the list of audio features used to train the network.
3. After training and creating each feature bank, are the weight frozen when training the whole network?
4. Regarding the use of one bank for each genre. Have you tried to use banks trained with all genres? I think the results in this scenario could bring interesting insights to the community.
5. Another question related to the one bank for each genre. Why not condition the bank to the music style?


**Limitations:**

Yes. The limitations and broad impact are well discussed in the conclusion section.

**Strengths And Weaknesses:**

- Strengths:
    - I enjoyed reading the paper, which is well written, and structured. It is easy to read, understand, and the tables and figures are illustrative.
    - Incorporating representation banks with learned features to leverage the learning of motion generation is a very interesting and well-formulated idea.
    - The experiments are done thoroughly. The user study, the experimental results with the baselines, and the ablation study are in great detail and defend the finding well.


- Weakness:
    - I have two minor concerns. First refers to the lack of motivation in the designing choices, such as the stage-wise strategy to train the network. Clear seems to be a good choice, but why is that in the case of dance motion generation.
    - Another possible weakness would be the limitation of requiring training the features bank for each genre. This limitation and designing decision could be better discussed in the paper since it limits the practical use of the approach.

In conclusion, the proposed method is very interesting, and I believe this is a solid paper that deserves to be communicated.

---

> ### Author Response · Authors · 2022-08-02
> **Response To Reviewer GKyZ (First Part)**
>
> Thanks to the reviewer for the constructive comments. We have carefully addressed your concerns and provided detailed responses for each review.
>
> **Q1: It is worth mentioning that the modules seem to be very useful in other applications where motion generation is required.**
>
> Re: Thank you for appreciating our work. To validate the wide applicability of our approach, we applied it to another dance generation method named Dance Revolution [10] and found it brings a clear performance boost. We also applied it to the more general 3D human motion prediction task and a preliminary attempt already improves the performance. These results validate the effectiveness and wide applicability of our method. Detailed analysis can be found in our response to Q1&Q2 for Reviewer 1dJr.
>
> **Q2. The minor concern refers to the lack of motivation in the designing choices, such as the stage-wise strategy to train the network. Clear seems to be a good choice, but why is that in the case of dance motion generation.**
>
> Re: The main reason for adopting stage-wise training is to ensure that the bank (learned in stage one) can accurately reconstruct the "GT motions". In contrast, if we adopt end-to-end training, the bank elements will be learned from the predicted noisy motions. During the rebuttal period, we tried the end-to-end training strategy and showed the detailed results in the table below. We can see that the stage-wise training strategy has clear advantages in terms of all metrics.
>
> |                             |         |         | Quality |          |            | Diversity |          | Alignment   |
> | --------------------------- | ------- | ------- | ------- | -------- | ---------- | --------- | -------- | ----------- |
> |                             | FID_k ↓ | FID_g ↓ | ∆Pose ↑ | ∆Trans ↑ | Freezing ↓ | Dist_k ↑  | Dist_g ↑ | BeatAlign ↑ |
> | Without Stage-wise Training | 29.37   | 16.75   | 1.52    | 1.29     | 33.4%      | 6.73      | 6.39     | 0.246       |
> | With Stage-wise Training    | 25.96   | 13.42   | 1.64    | 1.36     | 29.6%      | 7.68      | 6.59     | 0.249       |
>
> **Q3. Another possible weakness would be the limitation of requiring training the features bank for each genre. This limitation and designing decision could be better discussed in the paper since it limits the practical use of the approach.**
>
> Re: When professionals create a dance for a piece of music, they often already have something in mind about the dance genre. So it seems natural to leverage such genre information in choreography to achieve consistency in styles. Besides, considering that the number of genres is not large in practice, it doesn't have scalability issues. Nevertheless, we conducted the experiment of learning a single model for all genres and showed the results in the Table below. Our approach using genre-agnostic bank also brings notable improvement over the baseline.
>
> |                     |         |         | Quality |          |            | Diversity |          | Alignment   |
> | ------------------- | ------- | ------- | ------- | -------- | ---------- | --------- | -------- | ----------- |
> |                     | FID_k ↓ | FID_g ↓ | ∆Pose ↑ | ∆Trans ↑ | Freezing ↓ | Dist_k ↑  | Dist_g ↑ | BeatAlign ↑ |
> | Baseline            | 35.35   | 22.11   | 1.33    | 1.07     | 39.0%      | 5.94      | 6.18     | 0.241       |
> | Genre-agnostic Bank | 28.57   | 15.92   | 1.58    | 1.33     | 31.9%      | 7.29      | 6.42     | 0.247       |
> | Genre-specific Bank | 25.96   | 13.42   | 1.64    | 1.36     | 29.6%      | 7.68      | 6.59     | 0.249       |
>
> **Q4. Have you tried to train the network without applying the stage-wise strategy? What would be the performance of the methods?**
>
> Re: During the rebuttal period, we tried the end-to-end training strategy and detailed results have been presented in our response to Q2. We can see that the stage-wise training strategy has clear advantages in terms of all metrics. For detailed analysis, please refer to our response for Q2.
>
> **Q5. For the sake of reproducibility, kindly consider explicitly saying the list of audio features used to train the network.**
>
> Re: Thank you for the suggestion. We followed the practice in FACT [18] and extract a 35-dim music feature, which consists of 1-dim envelope, 20-dim MFCC, 12-dim chroma, 1-dim one-hot peaks and 1-dim one-hot beats.
>
> **Q6. After training and creating each feature bank, are the weight frozen when training the whole network?**
>
> Re: Yes, we pre-train the bank on GT motions first. Then, when we train the rest network components, we fix the bank elements. As a result, the bank serves as a prior probabilistic distribution where motions with noises will have small likelihood.

---

> ### Author Response · Authors · 2022-08-02
> **Response To Reviewer GKyZ (Second Part)**
>
> **Q7. Regarding the use of one bank for each genre. Have you tried to use banks trained with all genres? I think the results in this scenario could bring interesting insights to the community.**
>
> Re: Thank you for the suggestion. We have implemented this variant and detailed analysis can be found in our response to Q3.
>
> **Q8. Another question related to the one bank for each genre. Why not condition the bank to the music style?**
>
> Re: For dance generation task, existing data are collected and divided according to the dance genre. We just follow this common setting and construct Manifold Bank for each genre. We also consulted with professional dancers and learned that the music in the same style can be choreographed into dances with different genres. Constructing one bank for each dance genre can remove such ambiguity.

---

### Author Response · Authors · 2022-08-10
**Thanks to all reviewers for constructive comments**

Thanks to all reviewers for appreciating our work and providing constructive comments. We will provide experimental results and related analysis discussed during rebuttal process in the revised supplementary material to show broader use and generalizability of our proposed modules.

---

### Meta-Review · Area_Chair_waqv · 2022-08-27

**Recommendation:** Accept
**Confidence:** Certain

**Metareview:**

This paper proposes to fix common issues in motion generation by representing motion latents as combinations of discrete latent codewords learned by a VQ-VAE style approach. The idea is interesting and novel, and in the response period has been shown to potentially work for more than just dance generation, including a human trajectory prediction task.

While the idea might be quite well-motivated, reviewers all agreed that the writing does not quite do it justice in the submitted draft. I would recommend that the authors work on addressing those relevant reviewer comments for the next iteration (either camera-ready / future submission), aside from incorporating the new results into the draft.

**Award:**

No

---

### Decision · Program_Chairs · 2022-09-14

Accept